# Electromagnetic Interference Shielding Effectiveness of Direct-Grown-Carbon Nanotubes/Carbon and Glass Fiber-Reinforced Epoxy Matrix Composites

**DOI:** 10.3390/ma16072604

**Published:** 2023-03-24

**Authors:** Dong-Kyu Kim, Woong Han, Kwan-Woo Kim, Byung-Joo Kim

**Affiliations:** 1Industrialization Division, Korea Carbon Industry Promotion Agency, Jeonju 54852, Republic of Korea; 2Department of Carbon Materials and Fiber Engineering, Jeonbuk University, Jeonju 54896, Republic of Korea; 3Department of Advanced Materials and Chemical Engineering, Jeonju University, Jeonju 55069, Republic of Korea

**Keywords:** carbon fiber, glass fiber, carbon nanotube, composites, electromagnetic shielding

## Abstract

In this study, carbon nanotubes (CNTs) were grown under the same conditions as those of carbon fibers and glass fibers, and a comparative analysis was performed to confirm the potential of glass fibers with grown CNTs as electromagnetic interference (EMI) shielding materials. The CNTs were grown directly on the two fiber surfaces by a chemical vapor deposition process, with the aid of Ni particles loaded on them via a Ni-P plating process followed by heat treatment. The morphology and structural characteristics of the carbon and glass fibers with grown CNTs were analyzed using scanning electron microscopy–energy dispersive X-ray spectroscopy (SEM–EDS), X-ray diffraction (XRD), and X-ray photoelectron spectrometry (XPS), and the EMI shielding efficiency (EMI SE) of the directly grown CNT/carbon and glass fiber-reinforced epoxy matrix composites was determined using a vector-network analyzer. As the plating time increased, a plating layer serving as a catalyst formed on the fiber surface, confirming the growth of numerous nanowire-shaped CNTs. The average EMI SE_T_ values of the carbon fiber-reinforced plastic (CFRP) and glass fiber-reinforced plastic (GFRP) with grown CNTs maximized at approximately 81 and 40 dB, respectively. Carbon fibers with grown CNTs exhibited a significantly higher EMI SE_T_ value than the glass fiber-based sample, but the latter showed a higher EMI SE_T_ increase rate. This indicates that low-cost, high-quality EMI-shielding materials can be developed through the growth of CNTs on the surface of glass fibers.

## 1. Introduction

Emitted electromagnetic (EM) fields cause malfunctions in electronic devices and adverse human health problems such as headaches and dizziness. Thus, electromagnetic interference (EMI) generated by communication and broadcasting networks, power lines, and other electrical products has become a significant concern [1,2,3,4,5]. Due to the recent rapid development of the electronics industry, EMI shielding materials used to prevent such interference have been extensively investigated [6,7,8,9].

Metals such as copper, nickel, and stainless steel are considered the most effective EMI shielding materials [10,11,12]; however, their application is limited owing to high density and corrosion problems [13,14,15,16,17]. In recent years, research has been carried out on lightweight and anticorrosive polymer-based composites, wherein metal-plated fibers and carbon fillers are embedded in a polymer matrix, which can overcome the disadvantages of metal materials. Carbon nanotubes (CNTs) are excellent EMI shielding materials due to their low density, high surface area, and excellent electrical properties [18,19,20,21,22]. However, the utilization of CNTs is difficult because they easily aggregate owing to strong van der Waals interactions, resulting in poor dispersion [23,24]. Various physical and chemical methods such as ball-milling, ultrasonification, and plasma treatment have been used for CNTs dispersion. However, these methods require the use of high shear force or hazardous chemicals, which can damage the structure of CNTs and adversely affect the environment and human health [25,26,27,28,29,30].

These issues can be resolved through CNTs incorporation into fibers. When CNTs are grafted onto fibers, multifunctional composites with improved interfacial and electrical properties that allow EMI shielding of fibers can be fabricated by increasing the surface area of the fibers and imparting conductivity [31,32,33]. Impregnation, spraying, arc discharge, and chemical vapor deposition (CVD) are common methods for growing CNTs [34,35,36,37,38]. Among these technologies, CVD is an effective method to grow uniform, high-quality CNTs on fibers at a low cost and through a simple process [39,40].

Various industrial fibers, including carbon fiber, are used as reinforcements in the composite materials for enhancing EMI shielding properties [41,42]. Substantial research has been conducted in various fields because the growing of CNTs on fibers remarkably enhances the electrical properties, electromagnetic interference shielding efficiency (EMI SE), and interfacial properties [43,44,45]. However, because industrial fibers are subject to price restrictions when applied to products, the use of low-cost materials would lead to relatively diverse applications. Therefore, glass fiber, which is inexpensive and has high mechanical properties, is the best material for use in EMI shielding. To this end, conductivity must be imparted to glass fibers; some studies on coating metals such as nickel have been reported in this regard. However, few studies have focused on growing CNTs on the surface of glass fibers [46,47]. Such studies have mostly sought to improve electrical conductivity or the mechanical properties.

In this study, CNTs were grown using the CVD process under the same conditions as carbon and glass fibers to examine the potential of glass fibers with CNTs grown as EMI shielding materials. The surface morphologies of the carbon and glass fibers on which the CNTs were grown were characterized, and the potential of glass fiber as an EMI shielding material was demonstrated by comparing EMI SE using directly grown CNTs/carbon and glass fiber-reinforced epoxy matrix composites.

## 2. Materials and Methods

### 2.1. Materials

In this study, T300 woven carbon fiber (CO6644B, Toray, Tokyo, Japan) and woven E-glass fiber (1700WJ, Kunshan. Co., Kunshan, China) were used as reinforcing materials, and a difunctional epoxy diglycidyl ether of bisphenol-A (epoxy, YD-128, Kukdo Chem., Seoul, Republic of Korea) was used as the matrix, while 4,4′-Diaminodiphenylmethane (DDM, TCI Co., Tokyo, Japan) was used as the curing agent.

### 2.2. Nickel Plating and Post-Heat Treatment

Pretreatment with activation has been conducted to catalyze carbon and glass fibers. Carbon fibers and glass fibers were activated in a 0.38 M K_2_Cr_2_O_7_/4.5 M H_2_SO_4_ aqueous solution and refluxed in a water bath maintained at 60 °C for 2 h. Subsequently, SnCl_2_ and PdCl_2_ activating solutions were sequentially treated to form Sn/Pd nuclei for Ni reduction. Electroless Ni-P plating was performed by using nickel chloride (NiCl_2_·6H_2_O) and sodium hypophosphite (NaH_2_PO_2_·H_2_O) as the Ni source and reducing agent, respectively, at 85 ± 2 °C bath for 10, 30, 60, and 120 s. The post-heat treatment of nickel-plated carbon and glass fibers was performed at 600 °C for 30 min under N_2_ flow.

### 2.3. CNTs Grown on Fiber and Their Composites

For the growth of CNTs, the post-heat-treated fiber was loaded into a CVD chamber, heated up to 600 °C at 10 °C/min under a gas flow of Ar (500 cc/min) and H_2_ (200 cc/min), and the temperature was maintained for 30 min. Furthermore, C_2_H_4_ (200 cc/min) gas was introduced with Ar/H_2_ gas for an additional 10 min; the flow of C_2_H_4_ and H_2_ gases was terminated, followed by cooling to room temperature in an Ar atmosphere. The composite was fabricated by uniformly stacking three plies using the hand lay-up method and vacuum-packing at 160 °C under a pressure of 10 MPa for 1 h. Sample names are listed in Table 1.

### 2.4. Characterization

The surface morphologies of the carbon and glass fibers were studied before and after the growth of CNTs using scanning electron microscopy (SEM; S-4800, Hitachi, Japan). To prevent the sample from charging during SEM imaging, it was placed in a holder and coated with platinum nanoparticles at 4 mA for 150 s. The base pressure of the SEM chamber was approximately 5 × 10^−8^ Pa, and the acceleration voltage was 10 kV. For further material dispersion analysis, elemental mapping of the cross-section of the glass fiber was conducted using SEM–energy-dispersive X-ray spectroscopy (EDS).

The structural changes following plating, post-heat treatment, and CNT growth on the carbon and glass fibers were analyzed through wide-angle X-ray diffraction (XRD) using an EMPYREAN XRD instrument (PANalytical, Almelo, The Netherlands) equipped with a customized auto-mount and a Cu-*Kα* radiation source at 40 kV and 30 mA. Diffraction patterns were collected in the 2θ range of 30° to 90° at a scan speed of 2°/min. The spectral baseline was corrected, and the peaks were smoothed by connecting straight lines.

Furthermore, the surface chemical composition of the samples was analyzed through X-ray photoelectron spectroscopy (XPS, PHI 5000 Versa Probe II, ULVAC-PHI, Incorporated Company, Chigasaki, Japan). Unless otherwise specified, the X-ray anode was operated at >5 W, and the voltage was maintained at 5.0 kV. The energy resolution was fixed at 0.50 eV to ensure sufficient sensitivity. The base pressure of the analyzer chamber was ~5 × 10^−8^ Pa. Both the full-scan spectra (0 to 1200 eV) and narrow ones with a very high resolution for individual elements were recorded. Binding energies were calibrated with respect to the adventitious carbon peak (C_1s_: 284.6 eV). The high-resolution C_1s_, O_1s_, and Ni_2p_ peaks of the samples were deconvoluted using a Shirley-type baseline and an iterative least squared optimization algorithm. Furthermore, a curve-fitting procedure was carried out using a nonlinear least square curve-fitting program with a Gaussian–Lorentzian production function.

EM parameters were measured on a vector network analyzer (E5062A/EM2107A, Agilent Technologies, Santa Clara, CA, USA) with transmission-reflection mode according to ASTM D4935-89 in the range of 30–1500 MHz at room temperature. EMI SE was evaluated by measuring the attenuation or reduction of the EM wave; it was calculated and expressed in decibels (*dB*) by using the following equation [48]:(1)$$SE_{T}\left(dB\right)=10\log\frac{P_{1}}{P_{2}},$$
where *P*_1_ is the incident power and *P*_2_ is the transmitted power. The total *SE* (*SE_T_*) can be expressed as the sum of the reflection (*SE_R_*), absorption (*SE_A_*), and multiple reflection (*SE_MR_*) components, as follows [48]:(2)$$SE_{T}\left(dB\right)={SE}_{R}+{SE}_{A}+{SE}_{MR}$$

If the *SE_T_* value is greater than or equal to 15 dB, the effect of *SE_MR_* can be neglected, and the equation can be simplified as [48]:(3)$$SE_{T}\left(dB\right)\approx {SE}_{R}+{SE}_{A}$$

Furthermore, *SE_R_* and *SE_A_* can be calculated from the power coefficients, which are expressed as [48]:(4)$$SE_{R}\left(dB\right)=-10\log\left(1-R\right),$$
(5)$$SE_{A}\left(dB\right)=-10\log\left[1-\frac{A}{1-R}\right],$$
where *R*, *T*, and *A* represent the power coefficients of reflectance, transmittance, and absorbance, respectively. The *S* parameters, including *R*, *T*, and *A*, are determined from the incident power coefficients, and they can be expressed as [48]:(6)R+A+T=1,
$$R={\left|\left.S_{11}\right|\right.}^{2}={\left|\left.S_{22}\right|\right.}^{2},$$
$$T={\left|\left.S_{12}\right|\right.}^{2}={\left|\left.S_{21}\right|\right.}^{2},$$
where *S*_11_, *S*_22_, *S*_12_, and *S*_21_ correspond to the input reflection, output reflection, reverse transmission, and forward transmission, respectively. The measurement set-up consisted of a signal generator, specimen holder, and receiver. The composites were compressed under a pressure of ~10 MPa at 160 °C for 1 h using a hot press. The prepared sample size was 150 mm × 150 mm × 1.5 mm.

## 3. Results

### 3.1. Surface Morphology Analysis

The morphological changes of the carbon and glass fiber samples grown with CNTs according to the plating time are shown in Figure 1 and Figure 2. As shown in Figure 1, as the plating time increases, more Ni-P layers are formed on the carbon fiber surface; this confirms that Ni particles aggregate after post-heat treatment. The Ni-P layer thus formed served as a catalyst for the growth of CNTs. After CVD treatment, numerous nano-sized wire-like structures were formed; these nanowires formed a continuous network through curling and winding. As shown in Figure 2, as the plating time increases, a Ni-P plating layer is formed on the surface of the glass fiber, thus confirming CNT growth after CVD treatment. However, a pit-shaped surface was formed during post-heat treatment after Ni-P plating owing to volume reduction and the formation of a face-centered cubic (FCC) Ni crystal structure when the amorphous Ni-P plating layer is exposed to heat during the post-heat treatment. In the Ni-P plating layer, glass fibers were more uniformly formed than carbon fibers, and it is believed that the glass fibers with high polarity acted favorably toward the formation of Sn/Pd nuclei, resulting in a more uniform plating layer. This uniform plating layer acts as a catalyst for the growth of CNTs, and, thus, a larger amount of CNTs is formed compared to the carbon fiber surfaces.

Figure 3 presents the EDS mapping images of the cross-section of the Ni-P plated glass fiber before and after the growth of CNTs. After Ni-P plating, Ni was detected in the EDS image, which revealed the formation of a plating layer on the surface of the glass fiber. Furthermore, after the growth of CNTs, carbon, which was not observed previously, was detected. These results indicate the successful formation of the plating layer and subsequent growth of the CNTs on the glass fiber surface.

### 3.2. Crystalline Structure Analysis

Figure 4 shows the XRD patterns of the untreated, plated, post-heat treated, and CNTs-grown samples of the carbon and glass fibers. As shown in Figure 4a, the untreated carbon fiber has a C (002) peak at 2θ ≈ 26° and a small Ni-P amorphous peak at 2θ ≈ 45° after plating. After the post-heat treatment, the samples showed various peaks between 2θ ≈ 41° and 55° corresponding to the metallic Ni structure. This is because Ni is crystallized and precipitated after the post-heat treatment, thereby forming Ni_3_P, whose peak is observed in the XRD pattern. However, after CNT growth, the Ni peaks broadened again in comparison with those of the post-heat-treated carbon fibers. In general, as carbon fibers interact weakly with metals, when CNTs are grown using CVD, first, the hydrocarbon decomposes into carbon and hydrogen species upon contacting with hot metal nanoparticles on the fiber surface. Then, carbon diffuses under the metal and, finally, CNT forms beneath the metal particle, dislodging the entire metal particle off the carbon fiber surface [49]. Thus, the Ni peaks become prominent due to heat treatment after plating and also under the heat supplied during the process of growing CNTs. However, as CNTs grow and the Ni structure gradually collapses, the Ni peak broadens again. Additionally, the intensity of the C (002) peak decreased after plating and then increased again after CNT growth. This result confirms that the C (002) peak increased due to the growth of CNTs on the carbon fiber surface through the tip-growth mechanism. Similarly, in Figure 4b, a small Ni-P amorphous peak was observed at 2θ ≈ 45° after the glass fiber was plated. However, unlike carbon fibers, in glass fibers, sharp FCC Ni crystals were formed after the post-heat treatment. After the post-heat treatment, the thermodynamically unstable Ni-P plating layer forms a stable structure of FCC Ni crystals and body-centered tetragonal (BCT) Ni_3_P compounds. In an alloy with low P content, Ni precipitates first followed by Ni_3_P [50,51]. Therefore, in the case of the glass fiber, the sharp FCC Ni crystals were formed in the evenly and thickly coated Ni layer. Moreover, the intensity of the nickel peaks increased, and the formation of the C (002) peak after the CNTs grew. In the case of the glass fiber, highly polar interaction with the metal becomes the driving force, and the anchoring of metal particles to the glass fiber surface causes CNTs to grow on the metal surface, and not on the glass fiber surface. When CNTs are grown on glass fibers by CVD, first, the hydrocarbon decomposes around the metal surface; then, the dissolved carbon diffuses up to the metal surface; finally, the CNT grows on the metal particle. This base-growth model verifies that the intensity of the C (002) peak increases due to the growth of CNTs on the glass fiber surface. Additionally, during the growth of CNTs on the glass fiber surface, additional heat was supplied to the plating layer; this caused an increase in the intensity of the Ni peaks. The CNT growth mechanism is illustrated in Figure 5 [49].

### 3.3. Analysis of the Surface Characteristics

The chemical composition and surface properties of the post-heat-treated and CNT-grown glass fiber surfaces were investigated by XPS. The XPS profiles of the untreated, post-heat-treated, and CNT-grown glass fiber samples are shown in Figure 6. The spectra clearly show the peaks for carbon (C_1s_), oxygen (O_1s_), nickel (Ni_2p_), and silicon (Si_2p_) in the samples. The untreated glass fibers mainly show peaks for O_1s_ (binding energy (BE) = 531.3 eV), C_1s_ (BE = 284.3 eV), and Si_2p_ (BE = 101.9 eV). In the case of post-heat-treated samples, Ni_2p_ (BE = 855.8 eV) and Ni_3p_ (BE = 68.4 eV) peaks appeared, and although a plating layer was formed, the O_1s_ peak intensity did not decrease. We infer that the O_1s_ peak did not decrease due to the influence of the nickel oxide of the Ni plating layer, which was thickly formed on the surface of the glass fiber. In the XPS profile of the CNT-grown sample, the Ni_2p_, Ni_3p_ peaks, and O_1s_ peaks almost disappeared, and the intensity of the C_1s_ peak increased. This indicates that numerous CNTs were successfully grown on the plating layer.

The high-resolution XPS O_1s_ profiles of untreated and post-heat-treated glass fiber samples are shown in Figure 7. For the case of untreated glass fibers, the O_1s_ peak consists of Si-O and Si-O-Si peaks. However, NiO and Ni_2_O_3_ peaks also contributed to the O_1s_ peak after post-heat treatment following plating. This is due to the oxidation of Ni particles during the post-heat treatment after plating. The so-formed plating layer serves as a catalyst for the growth of CNTs, and the XPS O_1s_ peak almost disappeared after the growth of CNTs.

### 3.4. Electromagnetic Interference Shielding Efficiency Behavior

Generally, the frequency range of 0.3–1.5 GHz is associated with radio frequency (RF). RF is widely used in wireless communications and in other technologies, such as mobile communications, wireless optical communications, Bluetooth, Wi-Fi, and GPS. They are also used extensively in medical, industrial, and military applications. These frequency ranges are extremely important for research and applications aimed at addressing various technical problems. Figure 8a,b shows the power coefficients of A, R, and T calculated from S parameters, such as S_11_ and S_21_ collected under various conditions to determine the actual shielding ability of the carbon and glass fibers with surface-grown CNTs. In the case of carbon fibers, the R value of all samples after CNT growth is significantly higher than that of A. This result indicates that the amount of energy blocked by reflection is larger than that blocked by absorption. For glass fibers, there was little shielding effect up to HNCF-CNTs-30, because T was higher than A and R. However, as T decreased from HNCF-CNTs-60, it was confirmed that the shielding effect was improved, and R was rapidly improved as in the case of the carbon fibers. According to the EMI shielding theory [52], the interaction of the mobile charge carriers of a material leads to the reflection of EM waves, while the absorption of EM waves depends on the dielectric loss and/or magnetic loss of the material. Therefore, as CNTs were grown on the surface of the fiber, the electron mobility and electron density on the material surface were increased, which improved the interaction between the incident EM waves and electrons; consequently, the incident EM waves were significantly reflected. A, R, and T are quantitative parameters, while EMI SE is a relative parameter that is independent of the absolute power coefficient. Parameter A represents the ratio of the attenuated power of the EMI shielding material to the total incident power, but SE_A_ indicates the ability of the shielding material to disperse the EM waves. Therefore, the shielding mechanism is evaluated based on the contribution rates of SE_R_ and SE_A_ to SE_T_. Figure 8c,d show the EMI SE_T_ over the frequency range of 0.3–1.5 GHz of the carbon fiber-reinforced plastic (CFRP) and glass fiber-reinforced plastic (GFRP) with CNTs grown on the fiber surface as a function of the plating time. In Figure 8c, the CFRP with grown CNTs shows high EMI SE of >50 dB. As the plating time increases, many CNTs grow on the carbon fiber surface, improving the shielding behavior in the low-frequency region (30~600 MHz). However, a slight improvement in the shielding behavior was confirmed in the high-frequency region (600~1500 MHz) in relation to that in the low-frequency region. In Figure 8d, in the case of the GFRP with grown CNTs, GF-CNTs-10 and GF-CNTs-30 show less than 10 dB over all frequency ranges. However, as the plating time increases, the number of CNTs on the glass fiber surface increases, confirming that EMI SE improved up to ∼43 dB in the high-frequency region. These results indicate the possibility that glass fiber with grown CNTs can be used to replace carbon fiber as an EMI shielding material by changing the growth conditions of CNTs. Figure 8e,f shows the SE_T_, SE_A_, and SE_R_ values of the CFRP and GFRP with grown CNTs. The SE_T_, SE_A_, and SE_R_ values of both the carbon fiber and glass fiber composites tended to increase with increasing Ni plating time, and the SE_A_ value of all composites was higher than SE_R_, indicating that the shielding effect due to absorption dominated the SE_T_. This EMI shielding mechanism is schematically illustrated in Figure 9. First, as EM waves are incident on the surface of a shielding material, some of the EM waves are immediately reflected due to the impedance mismatch between the intrinsic impedance of the shielding material and the impedance of the propagation medium. This results in reflection losses, which occur owing to energy transmission through the boundary and the resulting energy loss. Thereafter, most of the EM waves that enter the shielding material are converted into heat owing to multiple reflections, resulting in absorption loss; that is, energy dissipation. As multi-layer composites have two or more layers, EMI shielding is achieved by the repetition of the EM wave absorption/reflection and EM wave penetration/re-absorption processes several times. The study findings suggest that the CNTs grown using a Ni-P layer, formed on the fiber surface via Ni-P plating, as the catalyst improve the EMI SE performance of the composite material.

## 4. Conclusions

In this study, CNTs were grown through the CVD process under the same conditions as carbon fibers (a conductor) and glass fibers (an insulator). A comparative analysis was performed to investigate the potential of glass fiber as an EMI shielding material. It was confirmed that a large amount of CNTs in the form of nanowires was formed on carbon and glass fibers as the plating treatment time increased. Glass fibers grew more CNTs than carbon fibers owing to the higher number of plating layers, which served as catalysts for CNT growth, formed on glass fibers with relatively high polarity. In terms of the EMI SE_T_, HNCF-CNTs-120 and HNGF-CNTs-120, which were prepared using the longest plating time, exhibited the highest EMI SE_T_ value in the frequency range of 0.3 GHz to 1.5 GHz. In the case of the carbon fiber, which is a conductor, the average EMI SE_T_ value over all frequency ranges tended to increase after the growth of CNTs, and all samples presented a high average EMI SE_T_ value of ≥70 dB. In the case of the glass fiber, which is an insulator, the HNGF-CNTs-30 sample exhibited a low EMI SE_T_ value; however, as the plating time increased, the average EMI SE_T_ value maximized to ~40 dB. Both carbon and glass fibers improved EMI SE_T_ after CNT growth; however, glass fiber showed a higher EMI SE_T_ increase rate compared to carbon fiber. These results suggest that low-cost and high-quality electromagnetic shielding materials can be developed via the growth of CNTs on the surface of glass fibers.

## Figures and Tables

**Figure 1 materials-16-02604-f001:**
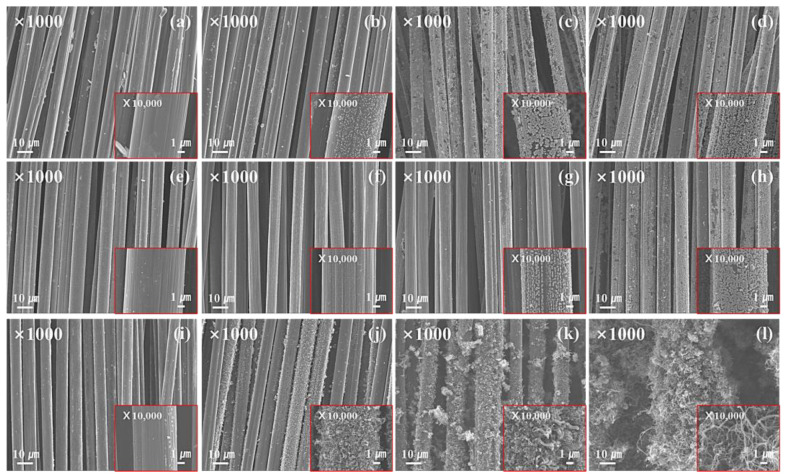
SEM images of the carbon fiber with CNTs grown on the surface; Ni-P-plated glass fiber; (**a**) NCF-10, (**b**) NCF-30, (**c**) NCF-60, (**d**) NCF-120, post-heat-treated glass fiber after Ni-P plating; (**e**) HNCF-10, (**f**) HNCF-30, (**g**) HNCF-60, (**h**) HNCF-120, glass fiber with CNTs grown on the surface after post-heat treatment; (**i**) HNCF-CNTs-10, (**j**) HNCF-CNTs-30, (**k**) HNCF-CNTs-60, (**l**) HNCF-CNTs-120.

**Figure 2 materials-16-02604-f002:**
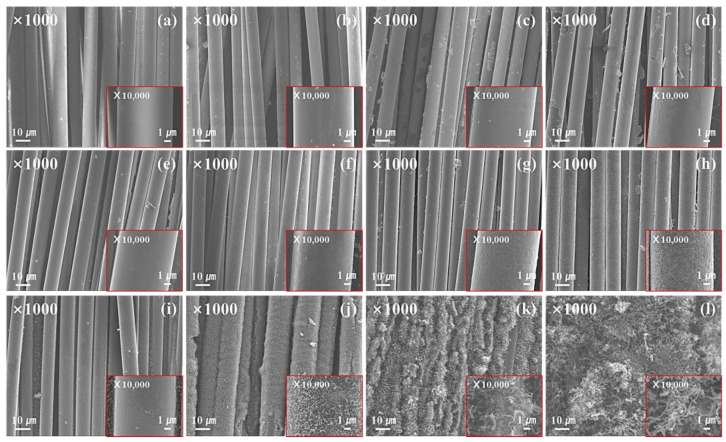
SEM images of the glass fiber with CNTs grown on the surface; Ni-P-plated glass fiber; (**a**) NGF-10, (**b**) NGF-30, (**c**) NGF-60, (**d**) NGF-120, (**e**) HNGF-10, post-heat-treated glass fiber after Ni-P plating; (**f**) HNGF-30, (**g**) HNGF-60, (**h**) HNGF-120, glass fiber with CNTs grown on the surface after post-heat treatment; (**i**) HNGF-CNTs-10, (**j**) HNGF-CNTs-30, (**k**) HNGF-CNTs-60, (**l**) HNGF-CNTs-120.

**Figure 3 materials-16-02604-f003:**
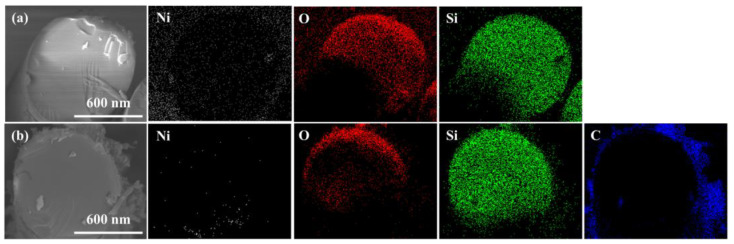
SEM–EDS mapping images showing the distribution of carbon, nitrogen, oxygen, silicon, and nickel in the cross-section of the glass fiber: (**a**) HNGF-120 and (**b**) HNGF-CNTs-120.

**Figure 4 materials-16-02604-f004:**
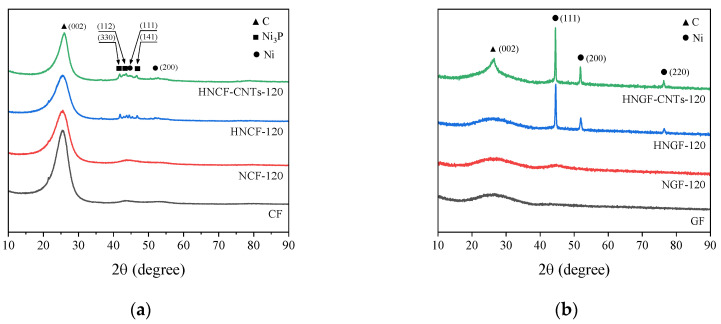
XRD patterns of the (**a**) carbon fibers with grown CNTs and (**b**) glass fibers with grown CNTs as a function of plating time.

**Figure 5 materials-16-02604-f005:**
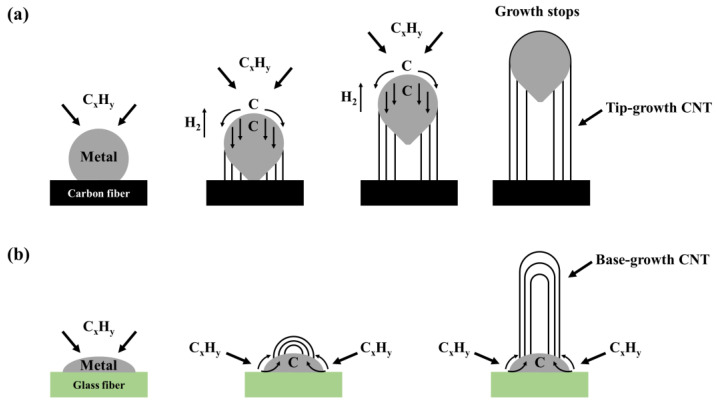
CNT growth mechanism: (**a**) tip-growth model and (**b**) base-growth model [49].

**Figure 6 materials-16-02604-f006:**
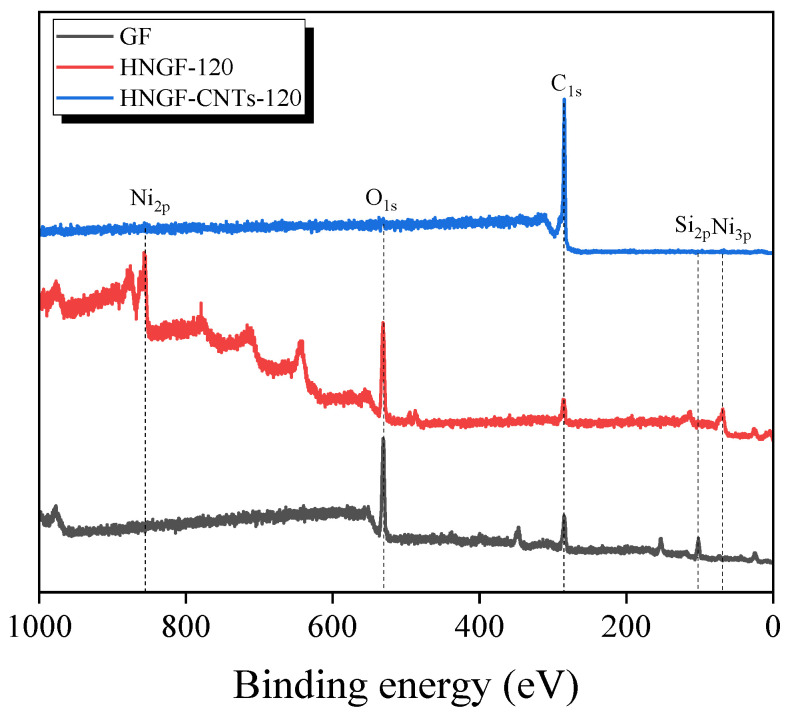
Survey-scan X-ray photoelectron spectra of the glass fibers modified by different treatments.

**Figure 7 materials-16-02604-f007:**
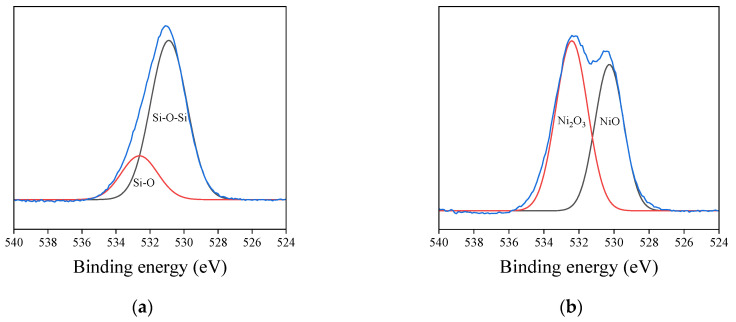
High-resolution XPS O_1s_ profiles of the (**a**) untreated glass fibers and (**b**) Ni-P plated glass fibers.

**Figure 8 materials-16-02604-f008:**
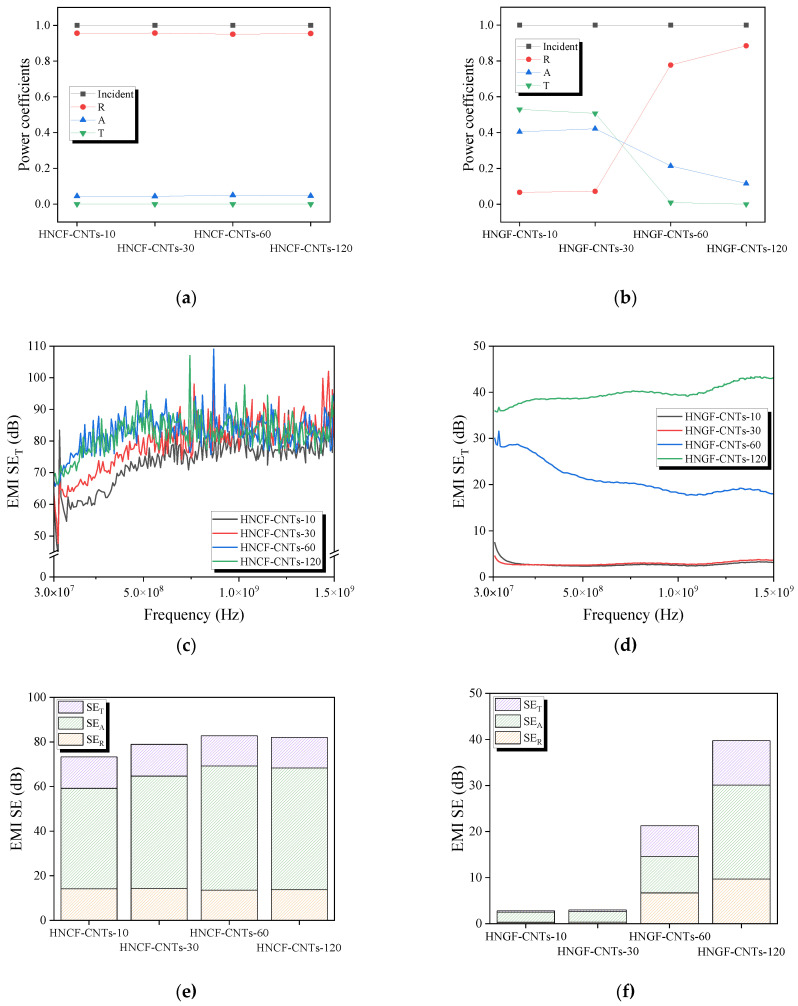
(**a**,**b**) Power coefficients of the CFRP and GFRP, respectively, with grown CNTs as a function of the plating time; (**c**,**d**) EMI SE_T_ of the CFRP and GFRP, respectively, with grown CNTs as a function of the plating time; (**e**,**f**) Contributions of SE_R_ and SE_A_ to the SE_T_ for CFRP and GFRP, respectively, with grown CNTs as a function of the plating time.

**Figure 9 materials-16-02604-f009:**
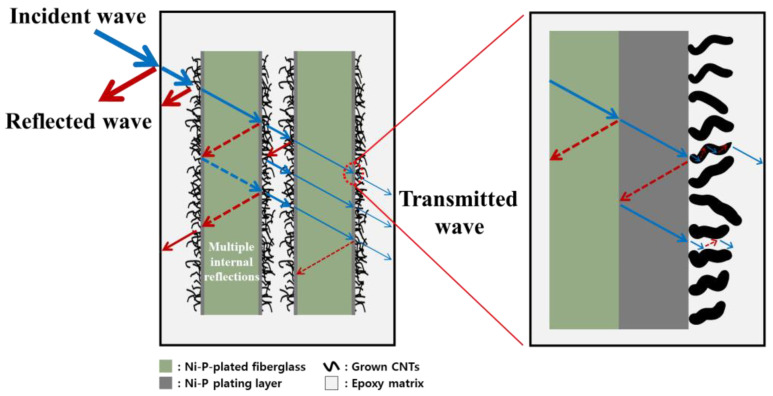
Schematic of the EMI shielding behavior of HNGF-CNTs/epoxy composites.

**Table 1 materials-16-02604-t001:** Sample names according to treatment conditions.

Sample Name	Treatment Conditions		
	Ni-P Plating	Post-Heat Treatment	CVD Treatment
	Time (s)	Temp. (°C)	Temp. (°C)
^a^NCF-10	10	—	—
^b^HNCF-30	30	600	—
HNCF-^c^CNTs-120	120	600	600
^d^NGF-10	10	—	—
^e^HNGF-30	30	600	—
HNGF-CNTs-120	120	600	600

^a^NCF: Ni-P plating on the carbon fiber; ^b^HNCF: post-heat treatment after Ni-P plating on carbon fiberl; ^c^CNTs: carbon nanotubes grown on the surface; ^d^NGF: Ni-P plating on the glass fiber; ^e^HNGF: post-heat treatment after Ni-P plating on glass fiber

## Data Availability

Data are contained within the article.

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
