# Peer review of "Electromagnetic Interference Shielding Effectiveness of Direct-Grown-Carbon Nanotubes/Carbon and Glass Fiber-Reinforced Epoxy Matrix Composites"

_materials, 2023, doi:10.3390/ma16072604_

Round 1

Reviewer 1 Report

the manuscript describe an interesting results for publication in Materials,

i suggest some minor revision before acceptance for publication.

-authors must detailed the obtained results in abstract.

-authors need to cited more previous researches in the filed of composites and other types of composites compared with the obtained composites in the paper,

i suggest these papers to be cited:

  1.  Derdar, H.; Mitchell, G.R.; Mateus, A.; Chaibedraa, S.; Elabed, Z.O.; Mahendra, V.S.; Cherifi, Z.; Bachari, K.; Chebout, R.; Meghabar, R.; Harrane, A.; Belbachir, M. Green Copolymers and Nanocomposites from Myrcene and Limonene Using Algerian Nano-Clay as Nano-Reinforcing Filler. Polymers 2022, 14, 5271. https://doi.org/10.3390/polym14235271.  

2. Derdar, H.; Mitchell, G.R.; Mahendra, V.S.; Benachour, M.; Haoue, S.; Cherifi, Z.; Bachari, K.; Harrane, A.; Meghabar, R. Green Nanocomposites from Rosin-Limonene Copolymer and Algerian Clay. Polymers 2020, 12, 1971. https://doi.org/10.3390/polym12091971.

-if it possible to confirm the structure and the morphology of the obtained composites by other methods such as :FT-IR and TEM analysis. Also, authors need to provide the study of thermal properties of the obtained products.

-Please, you need to change secrion 4 by conclusion and describe all the obtained results in conclusion secrion.

Reviewer 2 Report

This work focuses on the growth of CNTs as electromagnetic interference shielding efficiency (EMI-SE) materials.As the authors state, glass fiber with grown CNTs showed a higher EMI-SE increase rate than carbon fiber with grown CNTs; this indicates that low-cost, high-quality EMI-shielding materials can be developed through the growth of CNTs on the surface of glass fibers.

This work is quite interesting; Nevertheless some revisions are needed in order to publish this manuscript.

1. A few more references should be added regarding the use of CNTs as electromagnetic shielding materials.

2. I suggest the authors to present both S12 and S11 measurements in order to see if the EM shielding is due to transmission, reflection or absorption.

3. I kindly ask the authors to calculate the shielding efficiency (SE) dut to transmission, reflection and absorption, and present the data.

4. The authors should discuss more about the SE of their samples and Figure 5. in order to make any argument for Fig. 5 they have to calculate SET, SER and SEA and then the SEtotal.

6. The authors should mention why they are working in 30–1500 MHz. What is the technological interest in this range? A couple o references would be helpful. Why don;t they study 3G/5G etc?

7. Could the authors support their findings using numerical simulations or analytical calculations? This is also essential..

8. What is the effect of the geometry/size of the CNTs or the thickness of the samples?

Reviewer 3 Report

The current study of  “Electromagnetic interference shielding effectiveness of direct-2 grown-carbon nanotubes/carbon and glass fiber reinforced 3 epoxy matrix composites” is interesting for readers and scientists.

However, I have some comments to enhance the overall version of the manuscript:

-          You mentioned in the abstract that “Glass fiber with 20 grown CNTs showed a higher EMI-SE increase rate than carbon fiber with grown CNTs; this indicates that low-cost, high-quality EMI-shielding materials can be developed through the growth of 22 CNTs on the surface of glass fibers” may you support it with some of your real results in the abstract.

-          I see that the previous studies are missing in the introduction part, may you summarize the up-to-date listed materials for Electromagnetic interference shielding applications

-          For using the Epoxy here, why did you use epoxy here? I know that this sample has a high sound velocity and is used for many other applications, but how about you?

“Novel highly-sensitive heavy metals sensor-based 1D phononic crystal for NiCl2 detection

Springer, September 2022 Optical and Quantum Electronics 54(12), DOI: 10.1007/s11082-022-04212-7”

-          Do you have a relation between the acoustic (effect of sound waves) on these samples? Because sometimes electromagnetic waves are accompanied by a sound wave. This effect is so interesting at least for your future application

-          Your explanations in the results section are good, but can you please enhance it with some literature review related to your used materials

-          There are some English comments, please try to scan the manuscript again

Round 2

Reviewer 2 Report

The authors have revised their manuscript following most of my comments.

This manuscript could be published in its present form.

Reviewer 3 Report

Thank you for your report.